# A novel acidification mechanism for greatly enhanced oxygen supply to the fish retina

Christian Damsgaard[1][†][*], Henrik Lauridsen[2], Till S Harter[3], Garfield T Kwan[3], Jesper S Thomsen[4], Anette MD Funder[5], Claudiu T Supuran[6], Martin Tresguerres[3], Philip GD Matthews[1], Colin J Brauner[1]

[1]Department of Zoology, University of British Columbia, Vancouver, Canada; [2]Department of Clinical Medicine, Aarhus University, Aarhus, Denmark; [3]Scripps Institution of Oceanography, UC San Diego, La Jolla, United States; [4]Department of Biomedicine, Aarhus University, Aarhus, Denmark; [5]Department of Forensic Medicine, Aarhus University, Aarhus, Denmark; [6]Università degli Studi di Firenze, Neurofarba Department, Sezione di Scienze Farmaceutiche, Florence, Italy

**Abstract** Previously, we showed that the evolution of high acuity vision in fishes was directly associated with their unique pH-sensitive hemoglobins that allow $O_2$ to be delivered to the retina at $PO_2$s more than ten-fold that of arterial blood (Damsgaard et al., 2019). Here, we show strong evidence that vacuolar-type $H^+$-ATPase and plasma-accessible carbonic anhydrase in the vascular structure supplying the retina act together to acidify the red blood cell leading to $O_2$ secretion. In vivo data indicate that this pathway primarily affects the oxygenation of the inner retina involved in signal processing and transduction, and that the evolution of this pathway was tightly associated with the morphological expansion of the inner retina. We conclude that this mechanism for retinal oxygenation played a vital role in the adaptive evolution of vision in teleost fishes.

*For correspondence: cdamsg@zoology.ubc.ca

Present address: [†]Aarhus Institute of Advanced Studies & Section for Zoophysiology, Department of Biology, Aarhus University, Aarhus, Denmark

Competing interests: The authors declare that no competing interests exist.

## Introduction

The retina of vertebrates, containing the light-sensitive photoreceptors required for visual perception, has a very high metabolic rate that must be supported by an adequate supply of $O_2$ (*Linsenmeier and Braun, 1992*). To support such a high $O_2$ demand, most tissues possess a dense network of capillaries that minimize diffusion distances, facilitating the transfer of $O_2$ from the red blood cells (RBCs) to the respiring mitochondria. However, pigmented RBCs in capillaries on the vitreous side of the retina scatter incoming light and absorb photons before they reach the photoreceptors (*Buttery et al., 1991*; *Chase, 1982*; *Country, 2017*; *Damsgaard et al., 2019*; *Yu and Cringle, 2001*). The result may be an impairment of visual acuity, which may explain why only the choroidal vessels lining the back of the retina (also called the choriocapillaris) perfuse the retinae of most vertebrates (*Country, 2017*) (see *Figure 1* for retinal anatomy). The combination of a high $O_2$ demand and limited $O_2$ diffusion associated with this anatomical arrangement introduces a tradeoff between retinal $O_2$ supply and visual performance common to all vertebrates (*Damsgaard et al., 2019*). Thus, any evolutionary increase in retinal thickness to support longer photoreceptors for improved light sensitivity, more ganglion cells for improved visual acuity, and more synaptic connections for improved visual processing would limit $O_2$ flux due to the larger diffusion distance, necessitating compensatory mechanisms that ensure $O_2$ delivery by either decreasing diffusion distance or increasing the $O_2$ partial pressure ($PO_2$) gradient; the driving force for $O_2$ diffusion (*Bellhorn, 1997*; *Buttery et al., 1991*; *Chase, 1982*; *Country, 2017*; *Damsgaard et al., 2019*; *Yu et al., 2009*). Some vertebrates that evolved a thick retina, did so by lining the retina's inner (i.e. light facing) side with a

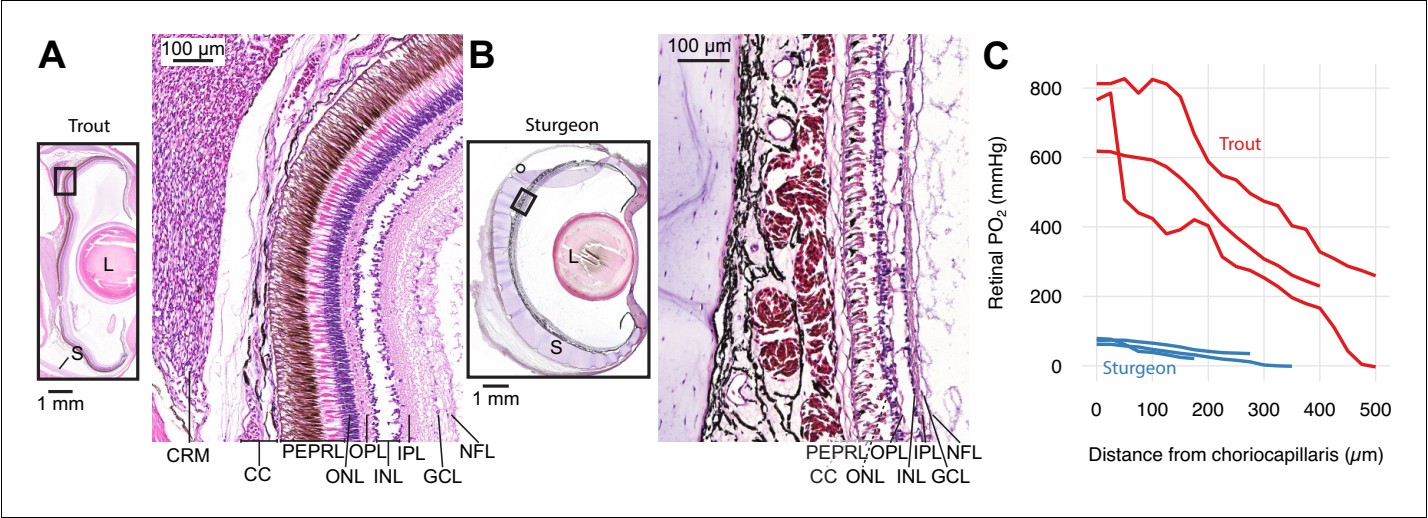

**Figure 1.** Retinal micro-anatomy and oxygen distribution in the retina of rainbow trout (*Oncorhynchus mykiss*) (A) and white sturgeon (*Acipenser transmontanus*) (B). Histology indicates a thick retina and presence of a choroid rete mirabile in the rainbow trout and a thin retina and absence of a choroid rete mirabile in the white sturgeon. PEPRL constitutes the outer retina, and ONL to NFL constitute the inner retina. Partial pressure of $O_2$ ($PO_2$) distribution in the retina of rainbow trout (red) and white sturgeon (blue) (C) showing that rainbow trout has a higher $PO_2$ (linear mixed-effects model, n = 6 [3 for each species], β ± SE = 390±75.8 mmHg, t = 5.14, p<0.001) and a steeper $PO_2$ gradient (linear mixed-effects model, n = 6 [3 for each species], β ± SE = −1.06 ± 0.0859 mmHg μm$^{-1}$, t = −12.4, p<0.001) when compared to white sturgeon. Abbreviations: CC, choriocapillaris; CRM, choroid rete mirabile; GCL, ganglion cell layer; INL, inner nuclear layer; IPL, inner plexiform layer; L, lens; NFL, nerve fiber layer; ONL, outer nuclear layer; OPL, outer plexiform layer; PEPRL, pigment epithelium and photoreceptor layer; S, sclera.

The online version of this article includes the following figure supplement(s) for figure 1:

**Figure supplement 1.** Vascular macro-anatomy of the eye of rainbow trout (*Oncorhynchus mykiss*).

high capillary density to halve the $O_2$ diffusion distance and thus promote $O_2$ delivery, while some mammals and a few teleosts have capillaries inside the retina (*Damsgaard et al., 2019*). In stark contrast, some teleost fishes evolved a unique solution to ensure sufficient $O_2$ delivery to their thick retina despite a complete lack of internal capillaries by using an exceptional system for $O_2$ supply, which delivers $O_2$ at a $PO_2$ far above that of the arterial blood (henceforth: $O_2$ secretion). A similar $O_2$ secretion mechanism was initially described in the teleost swim bladder, where blood acidification in the capillaries drives $O_2$ from hemoglobin (Hb) into the swim bladder lumen in order to regulate buoyancy against high hydrostatic pressures at depth (*Berenbrink et al., 2005*; *Steen, 1963a*). Later, $O_2$ secretion was also described in the teleost retina, where blood $PO_2$ can increase from around 85 mmHg in the central arteries to beyond 1300 mmHg in the retina of some fish species (*Randall and Cameron, 1973*; *Wittenberg and Wittenberg, 1962*). Oxygenated blood is supplied to the retina from the pseudobranch that drains into a counter-current rete (*Barnett, 1951*; *Copeland, 1980*; *Müller, 1841*; *Walls, 1942*), the choroid rete mirabile, which allows the high $PO_2$ to be magnified and localized behind the retina by permitting diffusion of $O_2$ from the venous to the arterial side of the rete. The evolutionary innovation of the choroid rete provided a high $O_2$ flux to the thick fish retina without obstructing the optical path. It likely enabled an increase in visual performance that may have facilitated niche expansions and novel feeding types and may have contributed to the explosive adaptive radiation of the teleosts, that today, represent half of all vertebrates (*Berenbrink et al., 2005*; *Randall et al., 2014*).

The hallmark of retinal $O_2$ secretion is the Root effect of teleost Hbs, where H$^+$-binding to Hb reduces not only its $O_2$ affinity but also its $O_2$ carrying capacity (*Pelster and Weber, 1991*; *Root, 1931*; *Waser and Heisler, 2005*; *Weber and Fago, 2004*). Thus, reductions in RBC pH lead to a significant right-shift of the $O_2$ equilibrium curve and, consequently, a pronounced increase in blood $PO_2$. However, the specific mechanisms responsible for acidifying the blood reaching the eye has remained largely elusive. While the pseudobranch slightly acidifies the blood before it reaches the eye, this acidification is not sufficient to activate the Root effect (*Bridges et al., 1998*; *Kern et al., 2002*; *Waser and Heisler, 2005*; *Waser and Heisler, 2004*), as otherwise, this would

lead to a maladaptive loss of $O_2$ to the surrounding water. Thus, an additional source of $H^+$-secretion of the blood draining the pseudobranch is required to activate the Root effect and induce retinal $O_2$ secretion. We propose that a key candidate for excreting $H^+$s from the endothelial cells into the lumen is the vacuolar-type $H^+$-ATPase (VHA). This multi-subunit enzyme uses the energy from ATP hydrolysis to transport $H^+$ across biological membranes (*Tresguerres, 2016*). VHA is expressed in cells within the pseudobranch and the swimbladder gas-gland (*Boesch et al., 2003*; *Kern et al., 2002*). However, $H^+$s do not rapidly cross the red blood cell membrane and must be combined with $HCO_3^-$ to form $CO_2$ ($HCO_3^-$ dehydration) that readily diffuses across the RBC membrane (*Brauner et al., 2019*). This dehydration reaction is slow relative to rates of blood capillary transit, so we also propose that a catalytic plasma-accessible carbonic anhydrase (paCA) is required to take advantage of VHA $H^+$-excretion for RBC acidification. Once $CO_2$ is within the RBC, it must rapidly be converted into $H^+$ by a RBC CA to activate the Root effect, as the vast majority of teleost Hbs are insensitive to direct $CO_2$-binding and respond solely to changes in RBC pH (*Damsgaard et al., 2014*; *Weber and Fago, 2004*). Thus, the first aim of this study was to determine whether the choroid rete mirabile is a $H^+$-secreting gland that can enhance retinal $O_2$ secretion in teleosts, and more specifically, whether this process relies on the combined action of paCA and VHA in the rete blood vessels.

Retinal $O_2$ secretion appears highly beneficial, as it magnifies the $O_2$ diffusion gradient and permitted the evolution of a thicker retina (*Damsgaard et al., 2019*). However, nothing is known about how this superior mode of $O_2$ delivery affected the function and evolution of the different retinal segments, which can shed insight into the co-evolutionary interactions between retinal $O_2$ delivery and specific visual modalities. The photoreceptor layer (PEPRL in *Figure 1AB*) constituting the outer retina lie directly adjacent to the choroid and are readily supplied with $O_2$ (*Country, 2017*; *Walls, 1942*) (see *Figure 1AB* for an overview of the retinal anatomy). In contrast, the components of the inner retina are positioned more distantly to the choroid (*Country, 2017*; *Walls, 1942*), and hence, changes in choroidal $O_2$ delivery would be expected to have profound effects on inner retinal function. Because this region includes the synaptic layers for lateral inhibition, the ganglion cells for signal integration, and the nerve fiber layer for signal transduction (ONL to NFL in *Figure 1AB*), visual performance could be greatly impaired by limited $O_2$ delivery. Thus, the secondary aim of this study was to assess the functional sensitivity of the inner and outer retina to changes in choroidal $O_2$ supply, by inhibiting the different components of the hypothesized oxygen secretion pathway. Moreover, to further compare those findings with the morphological evolution of the inner and outer retina during evolutionary gains and losses of retinal $O_2$ secretion based on existing literature data.

We designed this study to test three hypotheses: i) that the choroid rete mirabile acidifies the blood using VHA and paCA, which act together to permit $O_2$ secretion and thus greatly augment $O_2$ supply to the retina. ii) that the functional significance of $O_2$ secretion is greatest to the inner retina, which is therefore more sensitive to changes in choroidal $PO_2$ compared to the outer retina. iii) consequently, the evolutionary gains and losses of the choroid rete mirabile were associated with corresponding changes in the morphology of the inner, but not the outer, retina. To test these hypotheses, we first used immunohistochemical staining of choroid rete mirabile of rainbow trout to identify the presence and localization of VHA and paCA. Next, we measured retinal function and choroidal $PO_2$ simultaneously in an in vivo rainbow trout model during sequential inhibition of extra- and intracellular CAs, revealing both the mechanisms for retinal $O_2$ secretion in the choroid rete mirabile and the effects of choroidal $PO_2$ on retinal function. Finally, we compared these functional findings to literature data on retinal morphology from multiple species to investigate whether the most $PO_2$-sensitive parts of the retina display the most extensive morphological changes during evolutionary gains and losses of the choroid rete mirabile.

## Results

### Retinal oxygen distribution in species with and without a choroid rete mirabile

First, we confirmed the previous descriptions of a choroid rete mirabile and high $PO_2$ in the eye of rainbow trout (*Oncorhynchus mykiss*, Walbaum, 1792) and to compare the retinal $PO_2$ values to

those of white sturgeon (*Acipenser transmontanus*, Richardson, 1836) that is not expected to possess a choroid rete mirabile based on the absence of this structure in other members of the genus (*Berenbrink et al., 2005*; *Damsgaard et al., 2019*). Histological sectioning of both species' eyes confirmed the presence of a choroid rete mirabile in rainbow trout and its absence in white sturgeon (*Figure 1AB*). Micro-CT of a rainbow trout head filled with a radiopaque contrast agent further revealed the three-dimensional anatomy of vascular structures involved in $O_2$ secretion (*Figure 1— figure supplement 1*; *Supplementary file 1*). Maximal transverse thickness of the retina was 135 and 425 µm in sturgeon and trout, respectively. We measured transretinal $PO_2$ profiles in three individuals of the two species using an ultrathin $PO_2$-sensitive electrode. As predicted, we found a much higher $PO_2$ in the retina of rainbow trout compared to that of white sturgeon (linear mixed-effect model, n = 6 [3 for each species], $\beta \pm SE = 390 \pm 75.8$ mmHg, t = 5.14, p<0.001) with a $PO_2$ of $732 \pm 58.6$ and $71.22 \pm 4.94$ mmHg in the choroid of rainbow trout and white sturgeon, respectively. Further, these data showed a 6.2-fold steeper slope of the $PO_2$ profile in the trout retina compared to the sturgeon retina (linear mixed-effect model, n = 6 [3 for each species], $\beta \pm SE = -1.06 \pm 0.0859$ mmHg µm$^{-1}$, t = −12.4, p<0.001).

## Immunolocalization of vacuolar-type proton ATPase and carbonic anhydrase four in the choroid rete mirabile

To illuminate the pathways for retinal oxygen secretion, we tested for the presence, localization, and activities of VHA and paCA within the choroid rete mirabile of rainbow trout. Western blotting showed the presence of VHA and CA4 proteins in tissue homogenates of the choroid rete mirabile (*Figure 2—figure supplement 1*). To localize these proteins within the choroid rete mirabile, we used immunohistochemistry and confocal super-resolution microscopy on tissue sections, showing that CA4 and VHA were both present in endothelial cells of the choroid rete mirabile vasculature (*Figures 2* and *3*). To determine the subcellular location of these enzymes, we used the nucleus of the endothelial cell as an intracellular reference. Super-resolution images showed that both CA4 and VHA were present on the luminal side of the cell nucleus, likely in association with the plasma membrane of these endothelial cells (*Figures 2D* and *3D*). This finding is consistent with a plasma-accessible orientation of VHA and CA4, and these enzymes seem to line the entire vasculature of the choroid rete mirabile in rainbow trout. Lastly, we showed a robust enzymatic activity of VHA in choroid rete mirabile homogenates of $2.20 \pm 0.252$ µmol ADP h$^{-1}$ mg protein$^{-1}$. Together, these results confirm the presence, cellular orientation, and functional activity of VHA and CA4 in the choroid rete mirabile of rainbow trout.

## Pharmacological inhibition of oxygen secretion

Having identified the presence of VHA and CA4 in the luminal side of the choroid rete mirabile, we tested their functional role in facilitating retinal $O_2$ secretion in vivo. Therefore, we instrumented anesthetized rainbow trout with electrodes in the retina for measuring an electroretinogram and an ultra-thin $PO_2$-electrode in the choroid of the counter lateral eye to measure choroidal $PO_2$. A dorsal aortic catheter was used to deliver arterial injections of saline (control), C18 (a functionally membrane-impermeable CA inhibitor within the duration of the experiment), and acetazolamide (a much more membrane-permeable CA inhibitor that affects both extracellular and intracellular CA) (see *Figure 4—figure supplement 1* for experimental setup and timeline) (*Rummer et al., 2013*; *Scozzafava et al., 2000*; *Supuran, 2008*). Under control conditions, choroidal $PO_2$ was $517 \pm 95.3$ mmHg (mean ± standard error of mean, n = 6), which is significantly higher than the typical $PO_2$ values of 85 mmHg in the dorsal aortic blood of rainbow trout (*Randall and Cameron, 1973*). The control injection of saline did not affect choroidal $PO_2$ (linear mixed-effect model followed by Tukey posthoc analysis, n = 6, $\beta \pm SE = -32.2 \pm 84.7$ mmHg, z = −0.380, Holm-adjusted p=0.70, *Figure 4*). The subsequent injection of C18 decreased choroidal $PO_2$, to $226 \pm 99.5$ mmHg (n = 6, $\beta \pm SE = -259 \pm 84.7$ mmHg, z = −3.05, Holm-adjusted p=0.0068, *Figure 4*), verifying a role of paCA in facilitating Hb-$O_2$ unloading to the retina. The addition of acetazolamide further decreased choroidal $PO_2$, to $18.3 \pm 13.2$ mmHg (n = 6, $\beta \pm SE = -207 \pm 84.7$ mmHg, z = −2.45, Holm-adjusted p=0.0283, *Figure 4*), indicating an additional involvement of intracellular CAs in acidifying the red blood cell to enhance $O_2$ secretion to the retina. These results in combination with the

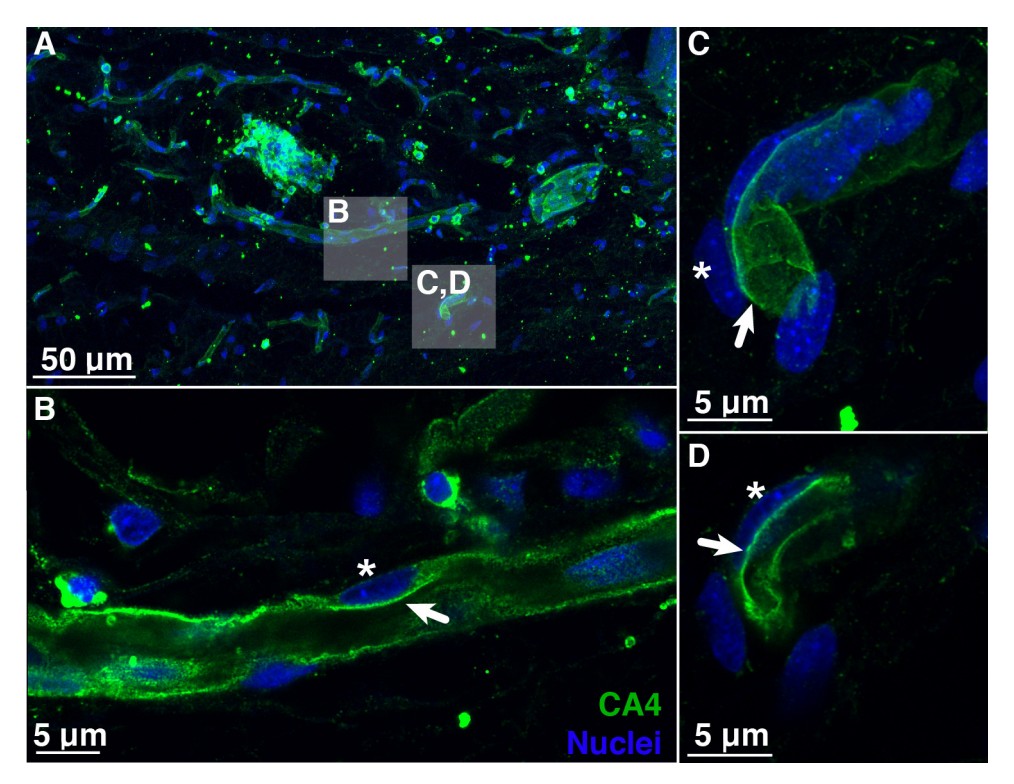

**Figure 2.** Blood vessels in the choroid rete mirabile of the rainbow trout eye, stained for the membrane-bound, carbonic anhydrase isoform 4 (CA4; green) and nuclei (blue). (**A**) Overview of a choroid rete mirabile section with inserts denoting the magnified vessels shown in panels B and C. (**B**) Magnified blood vessel shows CA4 staining along the entire length of the vessel. The staining pattern of a single endothelial cell is consistent with an apical localization of CA4 protein (arrow), whereas the nuclei of endothelial cells (see asterisk) are located basolaterally of the CA4 staining. (**C**) Another magnified blood vessel shows CA4 staining (arrow) surrounding the entire vessel lumen with the cell nuclei (asterisk) found on the outside. To further illustrate the luminal orientation of CA4, panel (**D**) represents a cross-section of the vessel shown in C, clearly showing that CA4 staining is not found within the vessel itself, but only surrounding the lumen. Detection of CA4 was with a polyclonal antibody raised against rainbow trout CA4 that has been described previously (*Georgalis et al., 2006*; *Gilmour et al., 2007*), and nuclei were visualized with DAPI.

The online version of this article includes the following figure supplement(s) for figure 2:

**Figure supplement 1.** Western blot analysis of choroid rete mirabile tissue homogenates.

immunohistochemical and biochemical data strongly support the hypothesis that VHA and paCA act together to facilitate $O_2$ secretion in the teleost retina.

## The role of oxygen secretion for retinal function

During the progressive inhibition of retinal $O_2$ secretion measured in the right eye, we quantified the functional implications for vision in the left eye by recording electroretinograms in response to pulses of 525 nm light (*Figure 4—figure supplement 1*). Each light stimulus event consisted of six 0.1 ms light pulses separated by 1 s intervals, with stimulations repeated every fifth minute. Under control conditions, rainbow trout exhibited the normal features of a vertebrate electroretinogram with distinct a- and b-waves (*Figure 5A*), representing the hyperpolarization of the photoreceptors and the depolarization of the ON-center bipolar cells within the inner retina, respectively. The amplitudes of the a- and b-waves of the electroretinogram ($V_a$ and $V_b$, respectively) displayed absolute reductions with reduced choroidal $PO_2$ (linear mixed-effect model using $log_{10}$-transformed $PO_2$, n = 4, $\beta \pm SE = 0.383 \pm 0.0565$, t = 6.79, p<0.001), but with a 2.1-fold greater effect on the b-wave

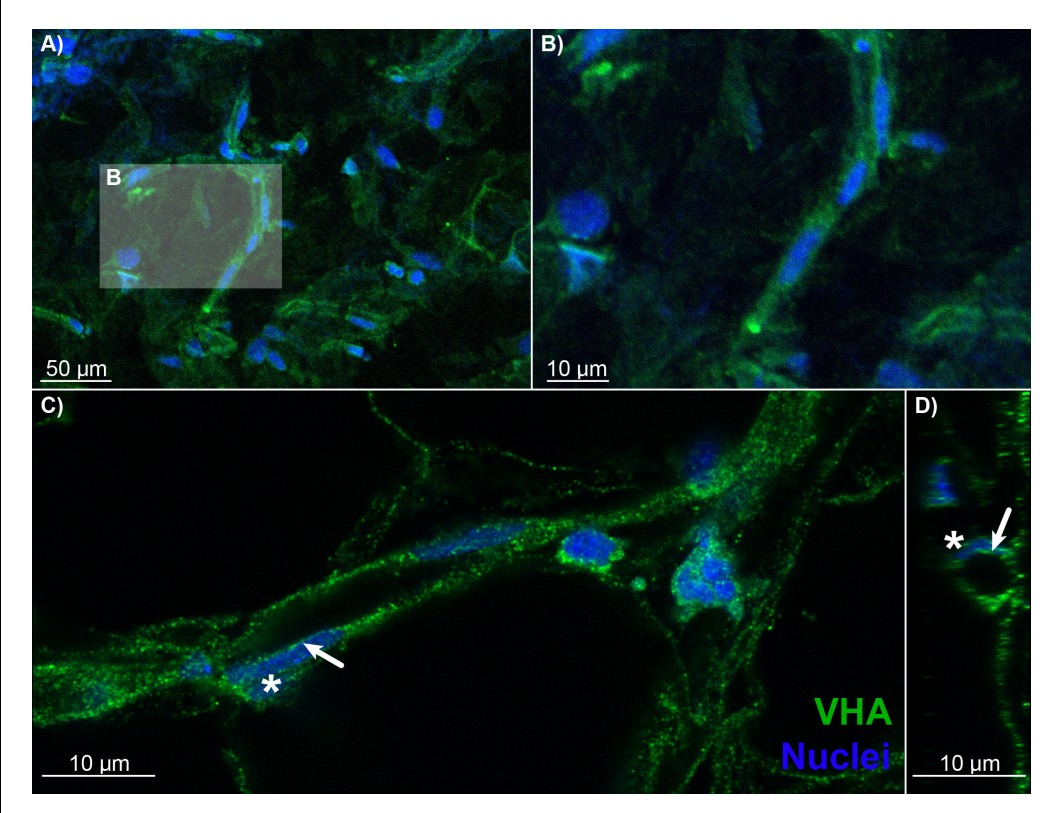

**Figure 3.** Blood vessels in the choroid rete mirabile of the rainbow trout eye, stained for the vacuolar-type proton ATPase (VHA; green) and nuclei (blue). (**A**) Overview of a choroid rete mirabile section with an insert denoting the magnified vessel section in (**B**) that shows a single blood vessel with VHA staining. (**C**) Another blood vessel shows the staining pattern for VHA along the entire length of the vessel walls. In a single endothelial cell, the staining pattern for VHA indicates an apical localization of the protein, whereas the nuclei of endothelial cells (asterisk) are located basolaterally of the VHA staining. To further illustrate the luminal orientation of VHA, panel (**D**) represents a cross-section of the vessel shown in C, further illustrating that VHA staining is not found within the vessel itself, but only surrounding the blood lumen. Detection of VHA was with a polyclonal antibody raised against the conserved β-subunit of VHA, and nuclei were visualized with DAPI.

compared to the a-wave (p<0.001). This finding supports the second hypothesis that the inner retina is more sensitive to changes in choroidal $PO_2$ compared to the outer retina.

Further, the data provided information about the temporal responsiveness of the inner and outer retina measured as the implicit time (IT) of the a- and b-waves, $IT_a$ and $IT_b$, respectively, quantified as the time from the light stimulus to the peak of the respective waves (*Figure 5A*). The data showed that $IT_a$ was impaired by reductions in choroidal $PO_2$ (linear mixed-effect model using $\log_{10}$-transformed $PO_2$ in mmHg, n = 4, β ± SE = −53.2 ± 7.93, t = −6.70, p<0.001). $IT_b$ was similarly affected by reductions in choroidal $PO_2$ (linear mixed-effect model using $\log_{10}$-transformed $PO_2$, n = 4, β ± SE = −52.8 ± 11.8, t = −4.47, p<0.001).

## The role of the oxygen secretion mechanism on the evolutionary dynamics of the inner and outer retina morphology

Using published data (*Damsgaard et al., 2019*), we compared the morphology of the inner and outer retina in species with and without a choroid rete mirabile (*Figure 6*). This analysis showed significantly thicker inner retina in species with a choroid rete mirabile (204 ± 92.6 μm [mean ± s.d.], n = 19) compared to species without (97.1 ± 37.0 μm [mean ± s.d.], n = 12) (phylogenetic analysis of variance simulation, p<0.001, n = 31). In contrast, the analysis did not show any differences in the thickness of the outer retina between species with and without a choroid rete mirabile (phylogenetic analysis of variance simulation, p=0.51, n = 31). Specifically, we identified significant differences in

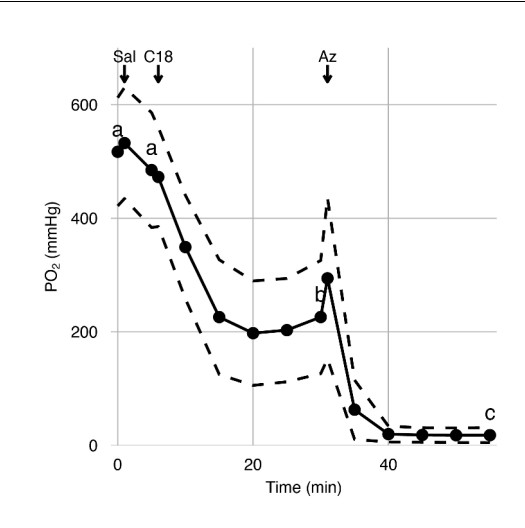

**Figure 4.** Effects of carbonic anhydrase inhibition on retinal oxygen secretion. Choroidal $PO_2$ was measured in anesthetized rainbow trout using an ultra-thin $PO_2$ electrode inserted into the choroid during sequential injections of saline, C18, or acetazolamide (Az) via a dorsal aorta catheter. The saline injection did not affect choroidal $PO_2$, but the injection of C18 and Az led to sequential reductions in choroidal $PO_2$, as indicated by letters that differ (linear mixed-effect model, p<0.001). Black circles connected by solid lines represent mean choroidal $PO_2$ at 15 time points, and dashed lines represent standard error of mean (n = 6).

The online version of this article includes the following figure supplement(s) for figure 4:

**Figure supplement 1.** Experimental protocol for in vivo inhibition of carbonic anhydrase.

retinal layer thickness between species with and without a choroid rete mirabile in the four layers positioned most distant to the choroid (i.e. the inner nuclear layer (p=0.0499, n = 31), the inner plexiform layer (p=0.00754, n = 31), the ganglion cell layer (p=0.104, n = 31), and the nerve fiber layer (p=0.00726, n = 31); *Figure 6—figure supplement 1*). These findings support the third hypothesis that the evolutionary gains and losses of the choroid rete mirabile during teleost radiation had large effects on the morphology of the inner retina, but not the outer retina.

## Discussion

Our data reveal a mechanism by which active acidification of the blood in the eyes of teleosts greatly enhances $O_2$ secretion to their largely avascular retinas. Here, VHA is predicted to pump $H^+$s into the lumen of the choroid rete mirabile, where paCA catalyzes the dehydration of $H^+$ and $HCO_3$ into $CO_2$ for RBC acidification. In addition, by studying the mechanism using various different techniques, we were able to independently verify this pathway using biochemical assays, immunohistochemistry and western blotting. Furthermore, inhibiting a key component of the mechanisms in vivo, the activity of paCA, enabled us to link active $O_2$ secretion to the functional performance of the retina, where in particular, inner retinal function was impaired in the absence of active $O_2$ secretion. This finding corroborates with phylogenetic evidence of a pronounced expansion of the inner retina upon the evolution of the $O_2$ secretion mechanism. This novel pathway solves the mystery of the primary route for RBC acidification to secrete $O_2$ into the retina, serving as a key exaptation for the adaptive increases in visual performance in early teleost evolution.

### The role of the choroid rete mirabile in localized blood acidification

The Root effect and the ability of teleost blood to desaturate during a localized acidosis to increase retinal $PO_2$ have been known for almost a century (*Root, 1931*; *Steen, 1963b*), and this desaturation is a critical component of the $O_2$ secretory mechanism (*Waser and Heisler, 2005*; *Wittenberg and Wittenberg, 1962*). However, the role of the choroid rete mirabile and the specific mechanisms in blood acidification was poorly understood and was primarily shaped by observations by *Fairbanks et al., 1974*. They showed that CA was present in choroid rete mirabile homogenates and hypothesized that *if* this CA was expressed in the venous side of the rete, and *if* it was also plasma-accessible, then this paCA would catalyze the hydration of $CO_2$ into $H^+$ and $HCO_3^-$ in the venous part of the rete, thereby facilitating the venous to arterial transport of $HCO_3^-$ within the rete. However, this paCA-catalyzed $CO_2$ hydration in isolation appears maladaptive as it would i) buffer the $H^+$s secreted into the blood in the pseudobranch, increasing the demands for anaerobic $CO_2$/ $H^+$-production by the retina to fuel $O_2$ secretion, ii) prevent counter-current multiplication of $CO_2$ in the rete, and iii) lead to a pronounced efflux of $H^+$s from the rete into the systemic circulation. In contrast to this previous framework for choroid rete mirabile function, our data strongly support an active $H^+$-secretory role of the choroid rete mirabile by identifying a VHA lining the endothelium (*Figure 7*). A co-localized paCA in the choroid rete mirabile dehydrates $H^+$ and $HCO_3$ into $CO_2$ that

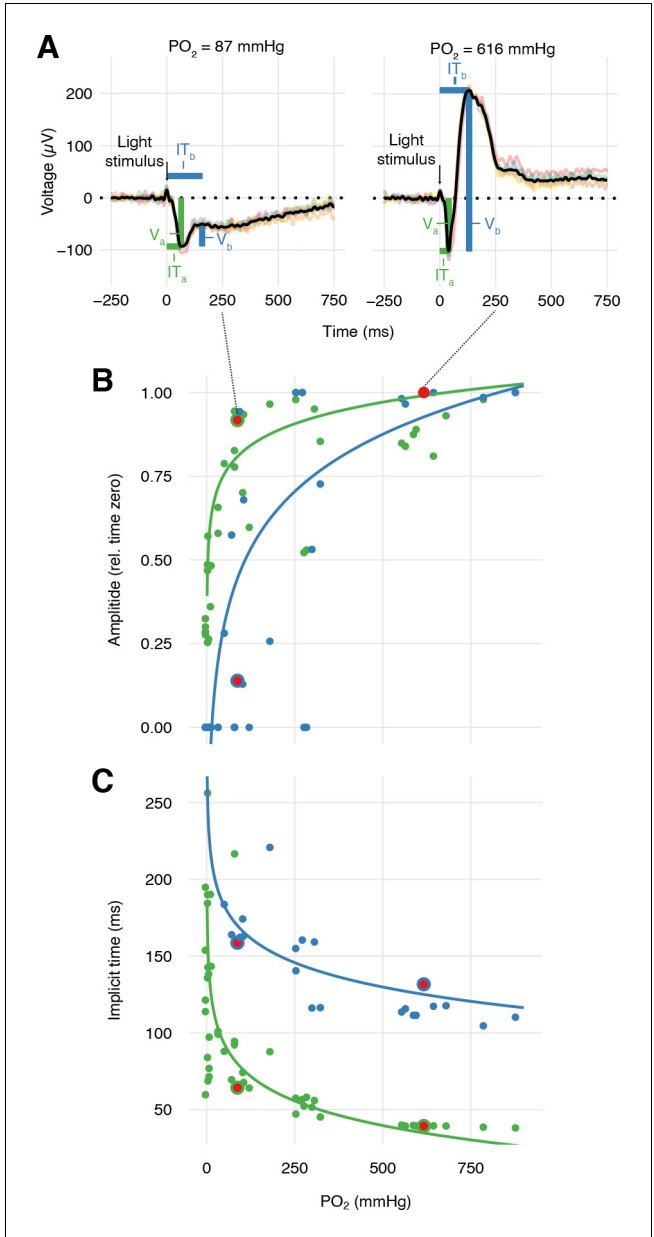

**Figure 5.** Effects of $PO_2$ on retinal function. (**A**) Representative sets of six electroretinograms (colored traces) at high and low $PO_2$ (right and left panels, respectively) measured after six 0.1 ms pulses of green light (black vertical arrow) and then averaged (black trace). The averaged electroretinograms were analyzed for the amplitude, V, and implicit time, IT, of the a- and b-wave (green and blue, respectively). Amplitudes (**B**) and implicit times (**C**) of the a- and b-wave in the electroretinogram of trout during progressive inhibition of retinal $O_2$ secretion by C18 and acetazolamide (n = 4). The properties of the electroretinograms in A are marked by red symbols. Solid lines are linear mixed-effect model fits to the data (all p<0.001, see text for statistics).

can diffuse into the RBCs (**Figure 7**). The presence of plasma-accessible CA4 is critical for this last step, as fish plasma has no carbonic anhydrase activity (**Henry et al., 1997**). By catalyzing the otherwise slow dehydration reaction, the catalysis couples $H^+$ secretion to the rapid production of $CO_2$ that, ultimately, drives the unloading of $O_2$ from Hb (**Figure 7**). This pathway is very significant for $O_2$ secretion, as inhibition of paCA with C18 resulted in a 60% reduction in choroidal $PO_2$. Thus, our data suggest that blood supplied from the pseudobranch is actively acidified within the choroid rete mirabile to activate the Root effect before the blood reaches the retina. This pathway does not

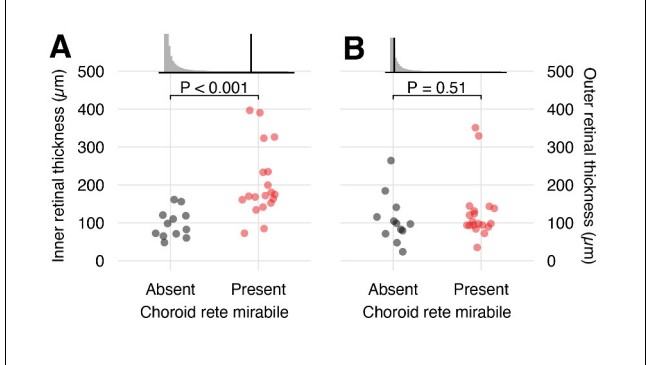

**Figure 6.** Effects of the presence of a choroid rete mirabile on the morphology of the inner - and outer retina. The transverse thickness of the inner- (**A**) and outer retina (**B**) in species with (red) and without (black) a choroid rete mirabile. Each dot represents species mean values obtained from the literature (***Damsgaard et al., 2019***). Effect sizes are visualized in the upper panels, showing the observed F-statistics (black vertical line) compared to a grey null-distributions of F-statistics from a phylogenetic analysis of variance simulation. This analysis showed a strong positive effect of the presence of a choroid rete mirabile on the inner retinal thickness (p<0.001, n = 31), but not on outer retinal thickness (p=0.51, n = 31).

The online version of this article includes the following figure supplement(s) for figure 6:

**Figure supplement 1.** Effects of the presence of a choroid rete mirabile on the retinal thickness of the individual retinal layers.

contradict pseudobranchial $H^+$-secretion, as proposed earlier, but instead indicates that the two tissues may act in concert to actively augment $O_2$ unloading from Hb within the rete.

While the current data strongly support the functional significance of a VHA – CA4 pathway for RBC acidification in the choroid rete mirabile, measurements of pH and $PCO_2$ in the afferent and efferent vessels of the choroid rete mirabile are necessary to provide conclusive evidence of a proton secretory function of the choroid rete. Unfortunately, pharmacological VHA inhibition in live animals is not possible due to the involvement of VHA in multiple processes, including neurotransmitter

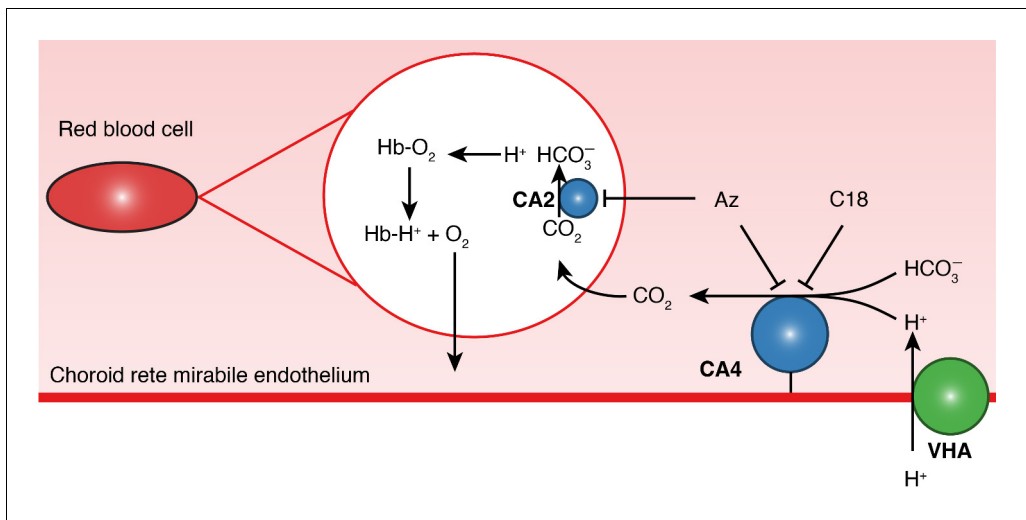

**Figure 7.** A proposed pathway for oxygen secretion in the choroid rete mirabile of teleost fishes. Endothelial $H^+$-secretion by vacuolar-type $H^+$-ATPase (VHA, green) drives $CO_2$ production facilitated by a plasma-accessible carbonic anhydrase (CA4, blue). This $CO_2$ diffuses across the red blood cell membrane and is then dehydrated by carbonic anhydrase 2 (CA2, blue), producing $H^+$s which bind to hemoglobin (Hb) and release $O_2$. C18 is a membrane-impermeable CA inhibitor that inhibits only the paCA (***Rummer et al., 2013***). Acetazolamide (Az) is a membrane-permeable CA inhibitor that rapidly inhibits both CA2 and CA4.

reuptake at synapses. Thus, further detailed studies on the role of VHA in acid excretion conducted ex vivo in isolated rete preparations are required.

Furthermore, our in vivo data suggest that the VHA – paCA pathway may not be the only mechanism for blood acidification to drive retinal $O_2$ secretion, as choroidal $PO_2$ remained above typical arterial values after paCA inhibition by C18. These results are indicative of pathways for anaerobic $CO_2$ production contributing to retinal $O_2$ secretion, where the produced $CO_2$ diffuses via the plasma and directly into the RBC without the need for paCA. Such pathways may include the pentose phosphate shunt, which has been suggested earlier to be involved in blood acidification in the retina (*Bridges et al., 1998*; *Pelster and Weber, 1991*).

## The effect of oxygen secretion on retinal function

Our data showed a strong relationship between choroidal $PO_2$ and retinal function with reduced amplitudes and increased implicit times of the a- and b-waves in the electroretinogram upon reductions in choroidal $PO_2$. Earlier studies showed pronounced changes in the features of the electroretinogram upon acetazolamide administration (*Fonner et al., 1973*). These changes were linked to retinal hypoxemia, as acetazolamide administration had been observed to reduce retinal $PO_2$ in separate studies (*Fairbanks et al., 1969*; *Fairbanks et al., 1974*). Our setup allowed for the simultaneous measurements of choroidal $PO_2$ and the electroretinogram during progressive reductions in choroidal $PO_2$ to add insight into the quantitative relationship between retinal hypoxemia and function.

Our data showed that the a-wave amplitude appears relatively unaffected by progressive retinal hypoxemia at high choroidal $PO_2$, but rapidly decreases with reductions in choroidal $PO_2$ below ~100 mmHg. These data indicate that photoreceptor function was only affected by retinal hypoxemia when choroidal $PO_2$ fell below normal arterial levels. In contrast, the b-wave amplitude showed higher sensitivity to a reduction in choroidal $PO_2$, suggesting a more pronounced dependence of the inner retinal layers to the mechanism of $O_2$ secretion by the choroid rete. These results show a tremendous sensitivity of the retina to reductions in choroidal $PO_2$ where retinal hypoxemia likely impairs the rate of oxidative phosphorylation in the mitochondria followed by the inability to maintain the transmembrane ionic gradients required for phototransduction (*Bickler and Buck, 2007*; *Boutilier and St-Pierre, 2000*). These differences in sensitivity of the inner and outer retina to reductions in choroidal $PO_2$ possibly stems from their different positions relative to the $O_2$ supply from the choroid rete: The photoreceptors comprising the outer retina lie directly adjacent to the choroid and experience the highest part of the $O_2$ diffusion gradient and are therefore less sensitive to reductions in choroidal $PO_2$. However, the synaptic layers comprising the inner retina are positioned further from the choroid and are, therefore, much more affected by reductions in choroidal $PO_2$. These results suggest that any phenotypic changes in the capacity to deliver $O_2$ at a high $PO_2$ in the retina have the most pronounced effects on the function of the inner retina. Such phenotypic changes may include ontogenetic changes in the expression of Hb isoforms with altered Root effect (*Bianchini, 2012*), an ATP-mediated reduction in the Root effect during hypoxia acclimation (*Weber and Lykkeboe, 1978*; *Wood and Johansen, 1972*), and an arterial desaturation during hypoxia or exercise making a smaller $O_2$ concentration available for $O_2$ secretion. Assessing the interactions between vision and environmental stressors that affect blood $O_2$ transport may provide novel insight into the respiratory mechanisms that affect fishes' visual performance.

## The choroid rete mirabile and the evolution of retinal functional morphology

Our in vivo identification of a higher $PO_2$ sensitivity of the inner- versus the outer retina suggests that any evolutionary change that increases choroidal $PO_2$ may act to offset an $O_2$ diffusion limitation on inner retinal $O_2$ supply and, thus, be a pre-requisite for the evolutionary expansion of the inner retina. Our interspecific comparison of retinal morphology in species with and without a choroid rete mirabile strongly supports this idea. Taken together, our in vivo and phylogenetic data independently suggests that the $O_2$ diffusion limitation to retinal thickness proposed in several studies (*Buttery et al., 1991*; *Chase, 1982*; *Country, 2017*; *Damsgaard et al., 2019*; *Yu and Cringle, 2001*) is mainly attributed to an $O_2$ constraint on the inner retinal morphology. A thicker inner retina can house more synaptic connections for intra-retinal detection of motion, changes in brightness,

and contrast generation in the inner plexiform layer before visual information is relayed via the ganglion cells to the brain (*Rodieck, 1998*). Thus, our findings imply that the origin of retinal $O_2$ secretion around 270 million years ago in a common teleost ancestor was associated with improved visual processing, which may have facilitated novel feeding strategies involving visual prey identification and tracking. Further, multiple groups have secondarily lost the $O_2$ secretory mechanism (*Berenbrink et al., 2005*; *Damsgaard et al., 2019*), which correspondingly suggest, that these groups have reduced capacities for visual processing.

Our retinal $PO_2$-values in white sturgeon and rainbow trout are consistent with expected values in the retina of species with and without a choroid rete mirabile, respectively (*Wittenberg and Wittenberg, 1962*). The evolutionary origin of $O_2$ secretion has mainly been discussed in the context of permitting the evolution of increased retinal thickness (*Berenbrink et al., 2005*; *Damsgaard et al., 2019*). However, the 10-fold higher choroidal $PO_2$ associated with the presence of a choroid rete mirabile was not associated with a proportional difference in retinal thickness between trout and sturgeon. Instead, the slope of the $PO_2$ profile was 6.2-fold higher in trout compared to sturgeon, which equals a proportionally higher $O_2$ consumption rate in the trout retina than the sturgeon retina (assuming equal diffusive capacity for $O_2$ in the retinae of the two species). Despite depending on two species only, this data set strongly suggests that the origin of $O_2$ secretion permitted not only the evolution of a thicker retina, but also the evolution of much higher metabolic rates of the retina. Increased retinal thickness is typically associated with a higher area density of photoreceptors (*Jeffery and Williams, 1994*; *Potier et al., 2017*; *Querubin et al., 2009*), which may explain the higher $O_2$ consumption rate, and which may provide an additional advantage to vision in terms on increased spatial resolution (*Damsgaard et al., 2019*).

## Summary

The choroid rete mirabile in the eye of teleost fishes allows for the generation of exceptionally high $PO_2$s in the retina and permitted the functional evolution of some of the most extreme retinal phenotypes among vertebrates with substantial expansions in the retinal layers associated with visual processing. Our identification of the VHA-paCA pathway for blood acidification to facilitate retinal $O_2$ secretion represents a novel physiological function of the choroid rete mirabile in addition to counter-current gas-exchange, which together permit greatly enhanced $O_2$ delivery to the retina. Our data further suggest that the origin of $O_2$ secretion not only permitted the evolution of a thicker retina, but also added fuel to the metabolic fire to visual perception. These significant advantages to vision possibly allowed teleost fishes to explore new food resources and colonize novel environments, which may have contributed to the extraordinary adaptive radiation of the teleost fishes, which today represent half of all vertebrates (*Berenbrink et al., 2005*; *Randall et al., 2014*).

# Materials and methods

## Animal housing

Rainbow trout (*Oncorhynchus mykiss*, Walbaum, 1792) and white sturgeon (*Acipenser transmontanus*, Richardson, 1836) were purchased as juveniles from Miracle Springs Inc (Mission, BC) and gifted from Vancouver Island University (Nanaimo, BC), respectively. The fish were housed in circular flow-through tanks with dechlorinated Vancouver tap water at 7˚C until experimentation in the Department of Zoology at the University of British Columbia (UBC). The fish were fed to satiation with commercial dry pellets three times weekly. All procedures were conducted according to the Canadian Council on Animal Care and approved by the UBC Animal Care Committee (Protocol no. A15-0266).

## Retinal $PO_2$ profiling

This experimental series was performed to assess baseline values for retinal $PO_2$ in rainbow trout, for comparison with those in white sturgeon which was not expected to possess retinal $O_2$ secretion based on the lack of a choroid rete mirabile in this genus (*Berenbrink et al., 2005*; *Damsgaard et al., 2019*). Individual animals were netted out of their holding tank, immersed into oxygenated dechlorinated tap water containing 50 mg $I^{-1}$ pH-neutralized MS-222 (3-aminobenzoic acid ethylester buffered with $HCO_3^-$). When branchial ventilation stopped, the animal was transferred to a surgical table with the left side up. The gills were constantly irrigated with oxygenated

dechlorinated tap water containing 40–50 mg l$^{-1}$ pH-neutralized MS-222 to maintain a surgical plane throughout the following experimental procedure, which lasted approximately 30 min.

Three rainbow trout (113 ± 10 g) and three sturgeon (875 ± 120 g) were used to measure transretinal $PO_2$-profiles. First, an 18G needle was used to make a small puncture in the cornea in the pupil's anterior midline inside the iris. This procedure allowed direct access to the vitreous humor and underlying retina unobstructed by the lens. Then, a thin $PO_2$ electrode with a 25 μm tip diameter (Unisense A/S, Aarhus, Denmark) was inserted into the eye through the corneal puncture using a motor-controlled micromanipulator (Zeiss, Germany). The electrode penetrated the retina in steps of 25 μm for 5 s each, while $PO_2$ was recorded with a temporal resolution of 100 ms on a computer using the manufacturer's software (Unisense Sensor Trace Suite). All $PO_2$ measurements were water vapor corrected using daily barometric pressure and relative humidity. After retinal $PO_2$ profiling, each animal was euthanized by adding additional MS-222 to 100 mg l$^{-1}$ and sampled as described below.

## Histology

Eyes from rainbow trout and white sturgeon were excised and fixed in 4% buffered paraformaldehyde. Eyes were dehydrated in an ethanol series (70%, 96%, 99.9%), embedded in paraffin, and cut in 5 μm intervals sections along the ventral-to-dorsal axis. All sections were stained with hematoxylin and eosin and imaged using bright-field microscopy with 4×, 10×, 20×, and 60× objectives. Maximal transverse thickness of the retina was measured as described in *Damsgaard et al., 2019*.

## Vascular casts and computed tomography scanning

To identify the three-dimensional anatomy of the vascular structures involved in $O_2$ secretion in the eye of rainbow trout, we generated a vascular cast of the eye by injecting a polymer-based radiopaque contrast agent (Microfil MV-122, Flow Tech Inc, USA), which was imaged by micro-CT.

One rainbow trout was anesthetized as described above and placed on a surgical table with the ventral side up. An incision was made to access the heart, and a polyethylene catheter (PE50) was inserted into the ventral aorta and secured with silk sutures. First, the animal was perfused with a 60 ml kg$^{-1}$ 0.9% NaCl solution containing 50 IU ml$^{-1}$ Na-heparin at a rate of 0.2 ml min$^{-1}$, using a syringe pump (Harvard Apparatus PHD 2000, USA). Then, the animal was perfused with 60 ml kg$^{-1}$ Microfil solution (compound:diluent:curing agent mix ratio was 1:3:0.4) at a rate of 0.1 ml min$^{-1}$, and the animal was placed on ice for 4 hr until the resin had hardened. Finally, the head was dissected and placed in 10 volumes of 4% buffered formaldehyde and stored at 4°C for imaging.

First, micro-CT was performed on the intact head region using an XtremeCT system (Scanco Medical AG, Brüttisellen, Switzerland) with 1500 projections/180° and an isotropic voxel size of 41 μm, an X-ray tube voltage of 59.4 kVp, an X-ray type current of 119 μA, and an integration time of 132 ms. Then, the right eye was carefully excised and scanned at a higher resolution using a Scanco Medical μCT 35 scanner in high-resolution mode (1000 projections/180°) with an isotropic voxel size of 10 μm, an X-ray tube voltage of 45 kVp, an X-ray tube current of 177 μA, and an integration time of 2400 ms.

Amira 5.6 (FEI, Visualization Sciences Group) software was used for anatomical model building.

## Perfusion clearing and fixation of tissues

Twelve rainbow trout were netted out of their holding tank individually and immersed into aerated dechlorinated tap water and euthanized with 100 mg l$^{-1}$ pH-neutralized MS222. When opercular ventilation stopped, the fish was transferred to a custom-made Styrofoam fish holder and injected in the caudal vein with 1 ml 0.9% NaCl containing 100 IU ml$^{-1}$ Na-heparin using a 23G hypodermic needle. Then, the bulbus was cannulated with a PE50 catheter, the tail cut off, and the animal was perfused with 0.9% NaCl containing 100 IU ml$^{-1}$ heparin via the catheter to remove all RBCs.

In four of those fish, the saline injection was followed by injection of 20 ml 4% paraformaldehyde in 0.9% NaCl. Both eyes were resected, fixed for 6 hr in 4% paraformaldehyde in 0.9% NaCl at 4°C, transferred to 50% ethanol overnight, and stored in 70% ethanol. Fixed tissues were shipped to the Scripps Institution of Oceanography (San Diego, CA) for immunohistochemistry.

In the remaining eight fish, the choroid rete mirabile was resected, flash-frozen in liquid nitrogen, and stored at −80°C for western blotting and to measure the enzymatic activity of VHA.

# Immunolocalization of vacuolar-type proton-ATPase and carbonic anhydrase four in the choroid rete mirabile

## Immunohistochemistry

Localization of CA4 in the *rete* was with a custom affinity purified rabbit polyclonal antibody raised against amino acids 57–74 (TRRTLPDERLTPFTFTGY; at 0.34 mg ml$^{-1}$; Abgent; San Diego, USA) of rainbow trout CA4 (GenBank AAR99330) and the antibody has been used and described in detail in previous studies on dogfish and rainbow trout (*Georgalis et al., 2006*; *Gilmour et al., 2007*). Other sections were immunolabeled with a rabbit anti-VHA antibody raised against the conserved β-sub-unit of VHA (AREEVPGRRGFPGY; at 3 mg ml$^{-1}$; GenScript, Piscataway USA), which has been used to localize VHA in teleost fish blood vessels and inner ear ionocytes (*Kwan et al., 2020*). For immu-nohistochemistry, fixed tissues were stepwise dehydrated in ethanol, cleared in SafeClear II (Fisher Scientific 23044192; Hampton, USA) and embedded in paraffin. Thick sections (20 µm) were cut on a KD-1508B microtome (KEDEE, Zhejiang, China) and mounted on Superfrost Plus (Fisher Scientific 12-550-15) microscope slides. After clearing the paraffin, the tissue sections were stepwise re-hydrated, and a hydrophobic barrier was drawn around each section (ImmEdge, Vector Labs NC9545623; Burlingame, USA). After a 5-min wash in phosphate buffered saline (PBS, Corning 46–013 CM; Conring, USA), about 100 µL of blocking buffer (PBS with 2% normal goat serum and 0.2% keyhole limpet hemocyanin) was pipetted onto each section and incubated in a humidified chamber for 1 hr at room temperature. Thereafter, the blocking buffer was decanted, and sections were incu-bated with the primary antibodies (CA4: 1:250; VHA: 1:500) in blocking buffer overnight at room temperature in a humidified chamber. On each slide, one of the tissue sections was used as a nega-tive control that was incubated with blocking buffer without a primary antibody. Detection of the pri-mary antibodies was with a goat anti-rabbit IgG conjugated to Alexa 488 (Invitrogen A32731, Carlsbad, USA). Sections were rinsed three times with PBS-T (PBS with 0.2% tween 20), then incu-bated with secondary antibody (1:500) and DAPI (1:1000) in a humidified chamber for 1 hr at room temperature. Sections were rinsed three times in PBS-T, and coverslips were mounted with Fluoro-Gel (EMS, Hatfield, USA) and sealed with nail polish. Sections were imaged with a Zeiss Observer Z1 inverted microscope with LSM 800 confocal and super-resolution Airyscan detector (Oberkochen, Germany) and using ZEN systems blue edition software (v2.6).

## Western blotting

The specificity of CA4 VHA antibodies was validated by Western blotting. Therefore, frozen choroid rete mirabile samples were first homogenized with a mortar and pestle under liquid nitrogen, and the powdered tissue was resuspended in homogenization buffer on ice (in mmol l$^{-1}$: 250 sucrose, 1 EDTA, 30, Tris at pH 7.5) with protease inhibitors DTT (Dithiothreitol; 1 mol l$^{-1}$), PMSF (phenylme-thylsulfonyl fluoride; 200 mmol l$^{-1}$) and BHH (benzamidine hydrochloride hydrate; 10 mmol l$^{-1}$). After that, rete tissues were homogenized further in a glass homogenizer and then centrifuged at 500 × g for 10 min to obtain a crude homogenate from the supernatant. Protein concentrations were measured with a Quick-start Bradford's assay (Bio-Rad, Hercules, USA) according to the manu-facturer's instructions, and bovine serum albumin was used for standards. Samples were diluted in 4 × Laemmli buffer (Bio-Rad 1610747) and heated at 70°C for 10 min. Aliquots of 5 µg of protein were loaded into each lane of a Mini-Protean TGX SDS-gel (Bio-Rad, 4568026) and separated at 60 V for 15 min followed by 200 V for 45 min. Proteins were then wet-transferred onto 0.2 µm Immun-Blot PVDF membranes (BioRad1620177) at 90 mA for 12 hr at 4°C. The transfer was assessed using total protein staining with 0.5% Ponceau-S in 1% acetic acid. Blots were rinsed with TBS-T and blocked with 10% skim milk powder in TBS-T on a shake table over night at 4°C. After that, the membrane was cut into sections that were probed with primary antibodies in blocking buffer (CA4 and VHA 1:2000) on a shaker table for 1 hr at room temperature. Additionally, peptide preabsorp-tion (1:5; antibody:peptide) served as a negative control for VHA. Protein size was assessed against a Precision Plus Protein Dual Color ladder (BioRad 1610374). All membranes were rinsed three times with TBS-T for 5 min at room temperature and were then incubated with a 1:5000 dilution of a goat anti-rabbit secondary antibody conjugated to horseradish peroxidase (IgG-HRP, Bio-Rad 1706515), on a shaker table for 1 hr at room temperature. Finally, membranes were rinsed with TBS-T, and pro-teins were visualized using a Clarity Western ECL chemiluminescent HRP substrate (Bio-Rad

1705061). Images were acquired in the Bio-Rad Universal III Hood and Image Lab software (Bio-Rad, v6.0.1).

## Enzymatic activity of vacuolar-type proton-ATPase in the choroid rete mirabile

To determine the activity of VHA in the choroid rete mirabile, we used a method in which ATP hydrolysis was linked to the oxidation of NADH via lactate dehydrogenase and pyruvate kinase and the reaction was followed spectrophotometrically (*McCormick, 1993*). The choroid rete mirabile were homogenized on a Precellys 24 tissue homogenizer (Bertin Technologies SAS, Montigny-le-Bretonneux, France; 5,000 rpm, 20 s, 4°C) using 1.0 mm ceria stabilized zirconium oxide beads (Next Advance, Averill Park, NY) in SEID buffer (150 mmol l$^{-1}$ sucrose, 10 mmol l$^{-1}$ EDTA, 50 mmol l$^{-1}$ imidazole, 0.1% sodium deoxycholate, pH 7.3). The homogenate was centrifuged (5,000 rpm, 1 min, 4°C), and the supernatant was used for the subsequent enzymatic assay. The ATPase activity was monitored in triplicate at 340 nm and 25°C in the absence or presence of 1 mmol l$^{-1}$ ouabain or 1 mmol l$^{-1}$ ouabain with 4 mmol l$^{-1}$ N-ethylmaleimide. The maximum velocity of the hydrolysis reaction was determined as the maximal slope of absorbance over time. Sample protein concentration was measured using Bradford's reagent and bovine serum albumin as protein standards. The maximum reaction velocity ($V_{max}$) was expressed as the rate of ADP hydrolysis per mg protein. The Na$^+$/K$^+$-ATPase activity was determined by subtracting control and ouabain $V_{max}$, and VHA activity was determined by subtracting $V_{max}$ during ouabain and ouabain+NEM inhibition.

## The role of carbonic anhydrase in retinal oxygen secretion in vivo

This experimental series was designed to identify and quantify the role of paCA on O$_2$ secretion (right eye) and retinal activity (left eye) by injecting intra- and extracellular CA inhibitors through a dorsal aorta catheter. The instrumentation consisted of three steps and lasted approximately 30 min in total per animal.

First, six rainbow trout (624 ± 259 g) were anesthetized as in *Retinal PO$_2$ profiling* and instrumented with an indwelling catheter in the dorsal aorta (PE50) that was secured to the dorsal lining of the mouth with a silk suture (*Soivio et al., 1975*). The fish was positioned in an upright position while still irrigating the gills with oxygenated water containing the anesthetic. During this procedure, the animal's position was fixed by stereotaxis, and its body was covered in wet paper towels to prevent cutaneous desiccation.

Second, a 23G needle was used to make a small puncture in the cornea into the anterior midline of the pupil of the left eye, and an Ag/AgCl electrode (0.008' diameter silver wire,~4 mm chlorided in sodium hypochlorite solution; Cat No. 786500, A-M Systems, WA) connected to the positive terminal of an amplified head stage was introduced into the vitreous humor of that eye using forceps (*Fonner et al., 1973*). A second identical Ag/AgCl reference electrode, connected to the negative terminal of the head stage, was inserted into the left nare, and two Ag/AgCl pellet electrodes (A-M systems) were used to ground the animal, one placed in the mouth and the other on the back. The potential between the two head stage electrodes was amplified using a differential amplifier (DP-311, Warner Instruments, Hamden, CT) set to 0.1 to 100 Hz (high-pass and low-pass, respectively) and gain set at 10 k. The amplifier's output was digitized using a Powerlab 8/35 DAQ analog to digital converter (ADInstruments, Bella Vista, NSW, Australia) and recorded continuously at 400 Hz on a laptop running LabChart software (ADIstruments). The green die of a tri-color LED (ASMT-MT00-00001, Broadcom Ltd., CA; peak wavelength 525 nm) was placed behind a semi-opaque white acrylic diffuser (2.7 mm thick, ST-S-401, Sumipex, Singapore) that was then positioned 5 cm away from the left eye to allow for light stimulations of the eye every fifth min to quantify retinal function (*Figure 4—figure supplement 1*). Six 0.1 ms-long pulses of green light spaced by 1 s were emitted from the LED by using the 0–10 V output of the PowerLab to drive a dimmable, buck-boost LED driver (A011-D-V-350 FlexBlock, LEDdynamics, Inc, VT). Electroretinogram data was not recorded in two of the six individuals due to electrode displacement during the insertion of the PO$_2$-sensitive electrode.

Third, an 18G needle was used to make a small puncture in the cornea at either the anterior or caudal midline of the pupil of the right eye. A PO$_2$ electrode with a 25-μm-thick tip was introduced into the eye as described in *Retinal PO$_2$ profiling* and was positioned in the choroid, as identified as

the tip position where $PO_2$ was maximal. Then, the room was darkened to allow the fish to dark adapt for 15 min, and all subsequent procedures were performed in complete darkness.

The experimental series was initiated by six light stimuli separated by 1 s at time zero and every 5 min throughout the 55 min protocol (*Figure 4—figure supplement 1*). At time 1 min, a bolus of 250 µl kg$^{-1}$ 0.9% NaCl solution with 50 IU ml$^{-1}$ heparin (saline) was injected into the dorsal aorta catheter. The effect of saline on the electroretinogram and choroidal $PO_2$ was assessed by comparing time points 0 and 5 min. At time 6 min, a bolus of 250 µl kg$^{-1}$ body mass of saline with DMSO (80:20) and 10 mmol l$^{-1}$ C18, a CA inhibitor with slow membrane permeability (*Scozzafava et al., 2000*; *Supuran, 2008*), was injected into the dorsal aorta catheter followed by a 250 µl saline bolus to clear the catheter (final blood C18 concentration of 0.2 mmol l$^{-1}$). A previous study has shown that this concentration of C18 will not significantly inhibit CA within rainbow trout RBCs over a 30 min time-frame (*Rummer et al., 2013*); therefore, for the duration of our protocol C18 was functionally membrane-impermeable. The effect of C18 on the electroretinogram and choroidal $PO_2$ was assessed every five mins over the next 25 min. At time 31 min, a bolus of 250 µl kg$^{-1}$ of a 27 mmol l$^{-1}$ acetazolamide, a highly membrane-permeable CA inhibitor (prepared in 50:50 saline and DMSO), was injected into the dorsal aorta catheter to achieve final blood acetazolamide concentration of 0.135 mmol l$^{-1}$. The effect of acetazolamide on the electroretinogram and choroidal $PO_2$ was assessed every five min over the next 25 min. All injections were followed by a 250 µl saline bolus to clear the catheter. The fish remained anesthetized throughout the following experimental procedure and was euthanized upon completion by immersion into water containing 100 mg l$^{-1}$ pH-neutralized MS-222.

## Calculations and statistical analyses

In the retinal $PO_2$ profiling, the effects of species and diffusion distance on retinal $PO_2$ was assessed using a linear mixed-effect model with species (rainbow trout or white sturgeon) and distance from the choroid as fixed effects and using individual fish as a random effect using the *lmer()*-function in the *lme4*-package for R (*Bates et al., 2015*). A likelihood ratio test was used to assess the effects of each fixed effect by comparing model fits with and without the fixed effect of interest.

The effects of injection of saline, C18, and acetazolamide on choroidal $PO_2$ were tested using a linear mixed-effect model using time as a fixed effect and individual fish as a random effect. To avoid problems with too many multiple comparisons, we compared choroidal $PO_2$ at time points 0, 5, 30, and 55 min only (see *Figure 4—figure supplement 1* for timeline). These four time points were included in the analysis to represent values for control (0 min), the effect of saline injection (5 min), the effect of C18 injection (30 mins), and the effect of acetazolamide injection (55 mins). Pairwise differences between time points were identified with a Tukey's honest significant difference test with a Holm correction.

The electroretinograms were analyzed for each of the twelve stimulus time points (see *Figure 4— figure supplement 1* for timeline). The electroretinogram was recorded for six stimuli, which were averaged to remove noise, and this averaged electroretinogram was analyzed for implicit time (IT) and amplitudes (V) of the a- and b-waves (see *Figure 5A* for graphical representation of the electroretinogram containing annotations for amplitudes and implicit times of the a- and b-waves). The a-wave amplitude ($V_a$) was found as the maximal hyperpolarization relative to the pre-stimulus potential (*Figure 5A*). The b-wave amplitude ($V_b$) was found as the maximal depolarization relative to $V_a$. IT of the two waves was determined as the time from the stimulus to reach $V_a$ and $V_b$. All measured electroretinograms had a distinct a-wave, but a distinct b-wave was lost at low $PO_2$s, so for those stimuli, $V_b$ was set to zero, and $IT_b$ was excluded from the subsequent analysis.

The effect of choroidal $PO_2$ on the implicit time of the a- and b-waves was tested using a linear mixed-effect model using log$_{10}$-transformed $PO_2$ and wave (a- or b-wave) as fixed effects and individual fish as a random effect. The choice of using log$_{10}$-transformed $PO_2$ rather than absolute $PO_2$ in the models was based on model selection using Akaike weight.

To test the effects of the presence of the choroid rete mirabile on the morphology of the inner and outer retina, we summarized 50,000 phylogenetic analysis of variance simulations (*Garland et al., 1993*) using *phylANOVA()*-function in the *phytools*-package in R (*Revell, 2012*) using published morphological data on the retina (*Damsgaard et al., 2019*) from 31 species, and a using a maximum clade credibility tree generated in *Damsgaard et al., 2020* from 100 Bayesian posterior

probability trees from *Rabosky et al., 2018*. The significance level was set to 0.05. All raw data and R scripts to analyze the data are available on https://figshare.com/articles/Source_data_1/12319763.

## Acknowledgements

We thank Jacelyn Shu and William K Milsom for technical assistance, Kathleen M Gilmour, Jonathan M Wilson, Steve F Perry, and Stephen D McCormick for contributing carbonic anhydrase antibodies, Dan Baker for the sturgeon, Kristian Beedholm for help with data analysis, and Jacelyn Shu for illustrating the rainbow trout in *Figure 4—figure supplement 1*.

## Additional information

### Funding

| Funder | Grant reference number | Author |
| --- | --- | --- |
| Carlsbergfondet | CF16-0713 | Christian Damsgaard |
| Carlsbergfondet | CF17-0195 | Christian Damsgaard |
| Carlsbergfondet | CF18-0658 | Christian Damsgaard |
| Carlsbergfondet | CF17-0778 | Henrik Lauridsen |
| National Science Foundation | IOS 1754994 | Martin Tresguerres |
| National Science Foundation | NSF Graduate Research Fellowship | Garfield T Kwan |
| Velux Fonden | | Henrik Lauridsen<br>Jesper S Thomsen |
| Natural Sciences and Engineering Research Council of Canada | 2018-04172 | Colin J Brauner |

The study design, data collection, and interpretation were not influenced by any funders.

### Author contributions

Christian Damsgaard, Conceptualization, Data curation, Formal analysis, Funding acquisition, Investigation, Visualization, Methodology, Writing - original draft, Project administration, Writing - review and editing; Henrik Lauridsen, Conceptualization, Resources, Formal analysis, Funding acquisition, Investigation, Visualization, Methodology, Writing - review and editing; Till S Harter, Conceptualization, Formal analysis, Investigation, Visualization, Methodology, Writing - review and editing; Garfield T Kwan, Formal analysis, Funding acquisition, Investigation, Visualization, Methodology, Writing - review and editing; Jesper S Thomsen, Resources, Funding acquisition, Investigation, Writing - review and editing; Anette MD Funder, Investigation, Writing - review and editing; Claudiu T Supuran, Resources, Writing - review and editing; Martin Tresguerres, Colin J Brauner, Conceptualization, Resources, Supervision, Funding acquisition, Methodology, Writing - review and editing; Philip GD Matthews, Conceptualization, Resources, Supervision, Investigation, Methodology, Writing - review and editing

### Author ORCIDs

Christian Damsgaard (iD) https://orcid.org/0000-0002-5722-4246
Henrik Lauridsen (iD) https://orcid.org/0000-0002-8833-4456
Till S Harter (iD) https://orcid.org/0000-0003-1712-1370
Garfield T Kwan (iD) https://orcid.org/0000-0001-9183-2731
Jesper S Thomsen (iD) http://orcid.org/0000-0001-9386-6679
Martin Tresguerres (iD) http://orcid.org/0000-0002-7090-9266
Philip GD Matthews (iD) https://orcid.org/0000-0003-0682-8522

## Ethics

Animal experimentation: All procedures were conducted according to the Canadian Council on Animal Care and approved by the UBC Animal Care Committee (Protocol no. A15-0266).

## Decision letter and Author response

Decision letter https://doi.org/10.7554/eLife.58995.sa1
Author response https://doi.org/10.7554/eLife.58995.sa2

---

# Additional files

## Supplementary files

• Supplementary file 1. Three-dimensional overview of the vasculature of the trout eye. Micro-CT-generated interactive overview of the ocular vasculature in a rainbow trout injected with a radiopaque contrast agent in the ventral aorta. Open the file in Adobe Acrobat Reader nine or higher and activate the 3D feature by clicking on the model. Then use the cursor to interact with the model or select pre-defined views similar to *Figure 1—figure supplement 1*.

• Transparent reporting form

## Data availability

All raw data and R scripts to analyze the data are available on Figshare (https://doi.org/10.6084/m9.figshare.12319763).

The following dataset was generated:

| Author(s) | Year | Dataset title | Dataset URL | Database and Identifier |
|---|---|---|---|---|
| Damsgaard C | 2020 | Source data 1 | https://figshare.com/articles/Source_data_1/12319763 | Figshare, 12319763 |

The following previously published dataset was used:

| Author(s) | Year | Dataset title | Dataset URL | Database and Identifier |
|---|---|---|---|---|
| Damsgaard C, Lauridsen H, Anette MD, Thomsen JS, Desvignes T, Crossley DA, Moller PR, Huong DTT, Phoung NT, Detrich HW, Bruel A, Wilkens H, Nyengaard JR, Berenbrink M, Bayley M, Wang T | 2019 | Retinaevolution | https://github.com/christiandamsgaard/Retinaevolution | Github, cc66d8d |

---

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
