## [Decision Letter]

Thank you for submitting your article "The choroid rete mirabile in the teleost eye is a proton-excreting gland that greatly enhances oxygenation of the retina" for consideration by *eLife*. Your article has been reviewed by three peer reviewers, and the evaluation has been overseen by a Reviewing Editor and Patricia Wittkopp as the Senior Editor. The following individuals involved in review of your submission have agreed to reveal their identity: Michael Country (Reviewer #2); Graham R Scott (Reviewer #3). Dr. Michael Country further pointed out that he had long discussions with you about future collaboration in the topic, but judged that he would be able to give an unbiased review.

The reviewers have discussed the reviews with one another and the Reviewing Editor has drafted this decision to help you prepare a revised submission.

We would like to draw your attention to changes in our revision policy that we have made in response to COVID-19 (https://elifesciences.org/articles/57162). Specifically, we are asking editors to accept without delay manuscripts, like yours, that they judge can stand as *eLife* papers without additional data, even if they feel that they would make the manuscript stronger. As one reviewer stated: "There are sufficient data at present for a publication without extra experiments being carried out although reference to future work directions could be made". Thus the revisions requested below only address clarity and presentation.

Summary:

In this study, Damsgaard et al. demonstrate that the choroid rete mirabile in the teleost eye contains V-type proton-pump and plasma-accessible carbonic anhydrase, which greatly enhances oxygenation of the retina when Root effect Hb is present. The results suggest that this mechanism has been important to enable inner retina to become thicker thus increasing visual acuity.

Essential revisions:

1) Although the manuscript has many new findings, definitive proof of the claims would have required (i) that the intracapillary pH and PCO_2_ were measured and (ii) that it had been shown that inhibition of the proton pump affected the choroid Po_2_. While it is acceptable that these measurements could not be done, their need should be stated in a future perspectives paragraph in the end of the Discussion. Because the involvement of proton pump in acidifying is not definitively shown, such involvement should be presented as intriguing and supported hypothesis, not a conclusive fact.

2) The existence of a metabolon between VHA and paCA is also a reasonable hypothesis that is worth proposing and discussing, but the authors should avoid definitive statements that one exists (of which there are many: Abstract, Results and Discussion) because the study has not provided convincing evidence for one. In order to be considered a metabolon, the two proteins would need to be shown to be in very close proximity to allow for substrate channeling. The data only show that both proteins can be found on the apical side of endothelial cells in the choroid rete. The proteins were separately identified in this general location, and they were not shown to be co-localized.

3) Do the authors have a trace of impaired ERG performance in low O2, for Figure 5?

---

## [Author Response]

Essential revisions:1) Although the manuscript has many new findings, definitive proof of the claims would have required (i) that the intracapillary pH and PCO_2_ were measured and (ii) that it had been shown that inhibition of the proton pump affected the choroid PO_2_. While it is acceptable that these measurements could not be done, their need should be stated in a future perspectives paragraph in the end of the Discussion. Because the involvement of proton pump in acidifying is not definitively shown, such involvement should be presented as intriguing and supported hypothesis, not a conclusive fact.

This is a good point. We have included a paragraph in the section “The role of the choroid rete mirabile in localized blood acidification”, where we discuss the limitations of the study, including the lack of intracapillary pH and PCO_2_. Here, we also point out that pharmacological VHA inhibition in vivo is not possible due to VHA´s involvement in multiple other physiological processes. Specifically, we write on: “While the current data strongly support the functional significance of a VHA – CA4 pathway for RBC acidification in the choroid rete mirabile, measurements of pH and PCO_2_ in the afferent and efferent vessels of the choroid rete mirabile are necessary to provide conclusive evidence of a proton secretory function of the choroid rete. […] Thus, further detailed studies on the role of VHA in acid excretion conducted ex vivo in isolated rete preparations are required.”

2) The existence of a metabolon between VHA and paCA is also a reasonable hypothesis that is worth proposing and discussing, but the authors should avoid definitive statements that one exists (of which there are many: Abstract, Results and Discussion) because the study has not provided convincing evidence for one. In order to be considered a metabolon, the two proteins would need to be shown to be in very close proximity to allow for substrate channeling. The data only show that both proteins can be found on the apical side of endothelial cells in the choroid rete. The proteins were separately identified in this general location, and they were not shown to be co-localized.

Thank you for pointing out that we did not show definitive evidence for a metabolon formed by VHA and CAIV. Unfortunately, the antibodies against VHA and CAIV were both raised in rabbits, which prevents co-localization studies on the same histological sections due to concerns about cross-reactivity of the secondary antibody (which is the same for the VHA and the CAIV antibodies). In response to this valid comment, we have rephrased all sentences that mention a metabolon to make it clear that we only showed that both, VHA and CAIV, are expressed on the luminal side of the choroid rete, and that they very likely are functionally linked for the purpose of red blood cell acidification (but without implying that they form a physically associated complex that mediates substrate channeling; i.e. metabolon).

3) Do the authors have a trace of impaired ERG performance in low O_2_, for Figure 5?

Yes, we have revised Figure 5 to include a ERG trace in low O_2_.